# *GhSTZ*-Mediated Suppression of Metabolic–Immune Coordination Compromises Cotton Defense Against Verticillium Wilt

**DOI:** 10.3390/plants14172638

**Published:** 2025-08-25

**Authors:** Guoshuai Zhang, Xinyu Zhu, Yanqing Bi, W. G. Dilantha Fernando, Xiaodi Lv, Jianfeng Lei, Peihong Dai, Yue Li

**Affiliations:** 1College of Life Science, Xinjiang Agricultural University, Nongda East Road, Urumqi 830001, China; 17325221793@163.com (G.Z.); 19922614126@163.com (X.Z.); b916228271@163.com (Y.B.); lxd479356@163.com (X.L.); kyleijianfeng@163.com (J.L.); peihong816@163.com (P.D.); 2Department of Plant Science, University of Manitoba, Winnipeg, MB R3T 2N2, Canada; dilantha.fernando@umanitoba.ca

**Keywords:** upland cotton, verticillium wilt, zinc finger proteins, VIGS, transcriptome

## Abstract

Verticillium wilt (VW), caused by *Verticillium dahliae*, poses a significant threat to global cotton production. Through analysis of public transcriptome databases, this study identified *GhSTZ*, a C2H2 zinc finger protein transcription factor gene, which was significantly induced by *V. dahliae*. Suppressing *GhSTZ* expression via virus-induced gene silencing significantly enhanced cotton resistance to VW. This resistance manifested as a 1.2-fold increase in lignin deposition, optimized ROS (reactive oxygen species) homeostasis, and a 1.3-fold elevation in glucose levels. Transcriptome analysis revealed 338 differentially expressed genes in *GhSTZ*-silenced plants, with 97 upregulated and 241 downregulated. Key downregulated genes included *PME* (pectin methylesterase) and *PG1-pec* (polygalacturonase) in the pentose phosphate pathway, while the key upregulated genes comprised *C4H* (cinnamate 4-hydroxylase) and *C3H* (p-coumarate 3-hydroxylase) in the phenylpropanoid biosynthesis pathway. Notably, in the plant–pathogen interaction signaling pathway, approximately half of the genes exhibited upregulated expression while the other half showed downregulation. Protein–protein interaction network analysis further revealed cooperative interaction between *PME* and the secoisolariciresinol dehydrogenase *SIRD*. This study is the first to elucidate *GhSTZ* as a negative regulator that compromises cotton disease resistance through a tripartite mechanism. These findings offer a novel approach to enhancing crop disease resistance by targeting the negative regulatory genes.

## 1. Introduction

Cotton (*Gossypium hirsutum*) is among the most important economic crops globally. Its fibers are essential raw materials for the textile industry, playing a vital role in the national economy [1]. However, cotton production faces threats from various pests and diseases. Notably, Verticillium wilt (VW), caused by the soil-borne pathogen *Verticillium dahliae*, is one of the most serious diseases reducing cotton yields [2]. The wilt pathogen invades cotton plants via the roots, subsequently colonizing and spreading within the vascular system [3]. This invasion leads to wilting, yellowing, and leaf drop, and in severe instances, causes total plant death, significantly reducing both yield and quality [4]. Currently, VW caused by *V. dahliae* is prevalent across all major cotton-producing countries, resulting in annual economic losses exceeding USD 1.2 billion in fungicide treatments alone while causing significant yield reduction and quality decline [5,6]. Breeding disease-resistant cotton varieties represents the most economically effective strategy for managing wilt. However, the mechanisms underlying cotton’s resistance to VW are complex, requiring coordinated regulation of multiple genes and metabolic pathways [6]. This complexity is partly attributed to the allotetraploid nature of the cotton genome, which complicates genetic regulation. Consequently, an in-depth analysis of the molecular mechanisms governing cotton resistance to VW is crucial for the development of resistant cultivars.

In recent years, considerable attention has been focused on the regulatory roles of transcription factors in plant disease resistance [7]. C2H2 zinc finger protein (ZFP) transcription factors represent one of the most prominent families of transcription factors in plants, playing a widespread role in plant growth, development, and responses to abiotic stress [8]. Research indicates that certain C2H2 ZFP transcription factors may negatively regulate plant immune responses by inhibiting pathways related to disease resistance. For instance, virus-induced gene silencing (VIGS) that downregulates *TaZFP8* expression enhances wheat resistance to *Puccinia striiformis f. sp. tritici* (*Pst*). In contrast, overexpression of *TaZFP8-5B* diminishes both wheat resistance to stripe rust and rice resistance to blast disease [9]. Notably, beyond direct regulation of immune genes, these transcription factors can also modulate disease resistance through reprogramming central carbon metabolism. In this context, the pentose phosphate pathway (PPP) serves as a critical metabolic nexus, significantly contributing to disease resistance by supplying precursors for biomolecule synthesis, participating in reactive oxygen species (ROS) metabolism, and maintaining redox balance [10,11]. Studies indicate that under low-temperature conditions, papaya can enhance respiratory metabolism within the PPP, boosting Nicotinamide Adenine Dinucleotide Phosphate Hydrogen (NADPH) levels Ato improve ROS scavenging capacity and thus increase the cold tolerance of papaya fruit [12]. Although the PPP’s role in disease resistance is documented, its integration into transcription factor-mediated networks remains poorly characterized.

Lignin significantly enhances plant resistance to fungi, bacteria, and viruses. It strengthens the physical barrier of plant cell walls, directly inhibits pathogen growth, and regulates the expression of disease-resistance genes [13,14,15,16]. Studies have demonstrated that silencing the *GhRac9* gene results in a significant decrease in lignin accumulation and content, along with reduced expression levels of genes involved in the lignin synthesis pathway [17]. Furthermore, GbCML45 and GbCML50 positively regulate cotton’s resistance to wilt disease by modulating the deposition pathways of lignin [18]. ROS play a crucial role in plant disease resistance by directly killing pathogens, enhancing oxidative cross-linking of cell walls, activating disease-resistance genes and signaling pathways (e.g., the salicylic acid pathway), and inducing hypersensitive necrosis responses, thereby constituting a multi-layered defense mechanism [19,20,21]. Compared to wild-type (WT) plants, silenced *GbPP2C80* plants exhibit increased ROS bursts and enhanced resistance to Fusarium wilt. Conversely, overexpression of *GbPP2C80* in Arabidopsis results in decreased ROS levels and weakened resistance compared to WT [22].

Glucose is a pivotal energy source and carbon skeleton provider in plants, engaging in diverse metabolic pathways via mechanisms encompassing energy supply, signal transduction, and metabolic network reconfiguration [23]. Research has demonstrated a close relationship between glucose metabolism and plant disease resistance, with glucose functioning as a signaling molecule to modulate the expression of defense-related genes in plants [24]. Glucose strengthens the physical barrier of plants against pathogens by facilitating the synthesis of lignin precursors and enhancing cell wall thickness [25]. For example, mutations in the rice UDP-glucose transporter gene *OsNST1* result in reduced cellulose content in cell walls, consequently diminishing resistance to the rice blast fungus [26]. Nevertheless, pathogens can exploit soluble sugars in plants as a nutrient source, giving rise to a competitive environment for sugar resources, thereby further complicating the regulatory network of plant disease resistance [27].

Employing a comprehensive analysis of public transcriptomic resources, we have identified *GhSTZ*, a pathogen-responsive C2H2 ZFP transcription factor gene demonstrating significant induction following challenge with *V. dahliae*, the fungal pathogen responsible for VW. To investigate the role of *GhSTZ* in cotton’s resistance to VW, we utilized VIGS technology to silence the *GhSTZ* gene and analyzed the resulting changes in cotton’s disease resistance. Furthermore, transcriptome sequencing was performed to evaluate how silencing the *GhSTZ* affects the gene expression profile of cotton. In addition, we measured levels of glucose, lignin, and reactive oxygen species to investigate the molecular mechanisms that regulate cotton’s resistance to VW. This study aims to elucidate the molecular mechanisms by which the *GhSTZ* transcription factor regulates cotton’s resistance to VW. The findings will provide a theoretical foundation for improving cotton disease resistance through targeted gene editing technologies that silence negative regulatory factors.

## 2. Results

### 2.1. GhSTZ Serves a Critical Role in Cotton Resistance to VW

To identify key regulatory genes involved in cotton’s response to *V. dahliae* infection, we analyzed genes differentially expressed before and after pathogen challenge using public transcriptome databases. Analysis revealed that *ZFP* gene expression underwent significant changes 24 h post-inoculation (hpi), with 10 genes significantly upregulated and 12 genes significantly downregulated (Figure 1a). However, the expression pattern shifted by 48 hpi, characterized by significant downregulation of 17 genes and relatively fewer upregulated genes (Figure 1b). To screen for core regulatory factors from these differentially expressed *ZFP* genes, we further constructed a Weighted Gene Co-expression Network Analysis (WGCNA) (Figure 1c). This WGCNA analysis unequivocally identified gene Gohir.D01G108400 (*GhSTZ*) as a central hub gene within the network. These results indicate that *GhSTZ* plays a pivotal core regulatory role in cotton’s defense against VW.

### 2.2. Bioinformatic Characterization and Expression Profiling of GhSTZ

Systematic bioinformatic analysis characterized the *GhSTZ* gene, localized on chromosome D01 of *G. hirsutum*, as an intron-free sequence encoding a 223-amino acid protein. The predicted molecular weight was 56.997 kDa with an isoelectric point of 5.09. Protein domain analysis identified two DNA-binding domains (DBDs) spanning residues 80–104 and 130–154.

Phylogenetic reconstruction using MEGA11 software with homologous sequences from NCBI categorized GhSTZ and its orthologs into five distinct clades (I–V), containing 13, 9, 12, 12, and 10 members, respectively (Figure 2a). GhSTZ exhibited the closest evolutionary relationship with protein TYH87758.1, suggesting functional conservation. Conserved motif analysis revealed shared motif composition and arrangement patterns across clades, with critical zinc finger domains (e.g., Motif 1 and Motif 3) overlaying DBD regions (Figure 2b). Domain prediction confirmed universal presence of C2H2 zinc finger domains among orthologs, demonstrating high intra- and inter-clade conservation essential for DNA-binding functionality (Figure 2c).

This study aims to investigate the function of the *GhSTZ* gene in upland cotton’s response to VW. We selected cotton samples from various tissues infected by the pathogen at 0 to 48 hpi to analyze the expression patterns of the *GhSTZ* gene under VW. Our results indicate that the *GhSTZ* gene is expressed in roots, stems, and leaves. In root tissues, gene expression initially increased and then decreased, peaking at 12 hpi with a level 8.03 times that of the control group before declining. In stem tissues, gene expression changes were more complex, reaching a maximum at 12 hpi (4.62 times that of the control group) and a minimum at 24 hpi (0.05 times that of the control group). Likewise, in leaf tissues, gene expression exhibited complexity, peaking at 24 hpi (7.84 times that of the control group) and reaching a low at 2 hpi (0.13 times that of the control group) (Figure 2d).

### 2.3. Silencing GhSTZ Enhances Cotton Resistance to VW

To elucidate the functional role of *GhSTZ* in cotton defense against VW, VIGS was employed to generate *GhSTZ*-suppressed plants (pTRV2::*GhSTZ*). The positive control pTRV2::*GhCLA1* plants exhibited characteristic albinism (Figure 3a), confirming the efficacy of the VIGS system. Reverse Transcription Quantitative PCR (RT-qPCR) analysis revealed a significant 65.1% reduction in *GhSTZ* transcript levels in pTRV2::*GhSTZ* plants compared to empty vector controls (pTRV2::*00*) (Figure 3b), validating successful gene silencing.

Twenty days post-inoculation with *V. dahliae*, *GhSTZ*-silenced plants displayed markedly enhanced disease resistance. Visual assessment revealed substantially reduced leaf wilting severity and diminished vascular browning in the stems of pTRV2::*GhSTZ* plants compared to controls (Figure 3c). Disease index quantification, as per the Chinese National Standard GB/T 22101.5-2009, confirmed significantly lower values in silenced plants (Figure 3d). Pathogen biomass assessment via *V. dahliae*-specific RT-qPCR demonstrated a pronounced decrease in relative fungal DNA abundance within stems of silenced plants when normalized to *GhUBQ7* (Figure 3e). Stem dissection further corroborated reduced vascular discoloration.

Collectively, these results demonstrate that *GhSTZ* functions as a negative regulator of VW resistance in cotton. Suppression of its expression significantly enhances host defense capacity.

### 2.4. Silencing GhSTZ Promotes Accumulation of Defense Compounds

To elucidate the physiological mechanism underlying the enhanced resistance conferred by *GhSTZ* silencing, we systematically quantified the accumulation levels of key defense compounds in *GhSTZ*-silenced plants (pTRV2::*GhSTZ*) compared to empty vector controls (pTRV2::*00*). Regarding ROS dynamics, 3,3′-diaminobenzidine (DAB) staining revealed significantly greater ROS accumulation (brown spots) in the leaves of silenced plants compared to controls, under both water-treated and *Vd*991-inoculated conditions (Figure 4a). Correspondingly, catalase (CAT) activity in silenced plants was 1.5-fold and 1.3-fold higher than in controls under non-inoculated and inoculated conditions, respectively (Figure 4b), indicating a concurrent enhancement in antioxidant capacity. Analysis of lignin deposition showed a markedly stronger phloroglucinol staining signal (red) in the vascular bundles of stems from silenced plants (Figure 4c). Lignin content was increased by 1.2-fold in silenced plants compared to controls under both non-inoculated and inoculated conditions (Figure 4d). Measurement of glucose levels revealed that leaf glucose content in silenced plants was 1.2-fold higher than in controls under non-inoculated conditions. Following pathogen inoculation, glucose levels were further elevated by 1.3-fold in silenced plants compared to inoculated controls (Figure 4e), providing ample carbon source support for defense responses.

Collectively, these changes in physiological indicators indicate that silencing *GhSTZ* establishes a more effective multi-layered defense barrier by coordinately regulating ROS homeostasis, cell wall fortification, and energy supply.

### 2.5. Differential Gene Expression Analysis and Validation

To elucidate the molecular mechanisms underlying enhanced resistance in *GhSTZ*-silenced plants, we performed transcriptome sequencing on stem tissues from silenced plants (pTRV2::*GhSTZ*) and empty vector controls (pTRV2::*00*) under *V. dahliae* stress. Differentially expressed genes (DEGs) analysis identified a total of 338 significant DEGs, comprising 97 upregulated and 241 downregulated genes (Figure 5a). This predominant downregulation pattern suggests *GhSTZ* primarily functions as a transcriptional repressor.

Gene Ontology (GO) enrichment analysis revealed significant enrichment of DEGs in three core functional categories: cellular components related to peripheral structures, notably the plasma membrane and extracellular region; molecular functions dominated by oxidoreductase activity and CAT activity; and biological processes primarily involved in oxidation-reduction reactions (Figure 5b). These findings are highly consistent with the observed phenotypes of ROS homeostasis remodeling and enhanced lignin deposition.

Kyoto Encyclopedia of Genes and Genomes (KEGG) pathway enrichment analysis further highlighted the pentose and glucuronate interconversions pathway and the phenylpropanoid biosynthesis pathway as the two most significantly enriched metabolic pathways (Figure 5c). These pathways are functionally linked to glucose metabolism/NADPH supply (for defense) and lignin synthesis, respectively. The plant hormone signal transduction pathway emerged as the most significantly enriched immune signaling pathway.

Protein–protein interaction (PPI) network analysis demonstrated that proteins encoded by DEGs formed a significant interaction network. Key hub genes within this network included the PPP rate-limiting enzyme *PME* (Ghi_D06G03566, Ghi_A07G00931, Ghi_A01G08156) and the phenylpropanoid pathway core enzyme *SIRD* (Ghi_D01G10786). Notably, close interactions were identified between *PME* genes and *SIRD* genes (Figure 5d).

Collectively, these multi-omics findings indicate that silencing *GhSTZ* triggers a global transcriptional reprogramming of the cotton defense response by relieving repression on key genes within the pentose phosphate pathway, phenylpropanoid synthesis pathway, and immune recognition pathways.

### 2.6. GhSTZ Regulates Expression of Key Genes in Defense-Related Metabolic Pathways

Silencing of *GhSTZ* induced differential expression patterns of key enzyme genes within the PPP and phenylpropanoid biosynthesis pathway. In the pentose phosphate pathway, concomitant with reduced *GhSTZ* expression, the *PME* (pectin methylesterase) gene was significantly downregulated (Figure 6a). This repression inhibits the elongation process, converting monomeric 1,4-α-D-galacturonide into linear polygalacturonate [(1,4-α-D-galacturonide)ₙ]. Concurrently, the expression of the *PG1-pec* (polygalacturonase) gene was reduced, hindering the degradation of digalacturonate into D-galacturonate. In contrast, two homologous *PLY* (pectate lyase) genes exhibited significant upregulation (Figure 6b).

The phenylpropanoid pathway displayed a distinct regulatory pattern. Expression of the *C4H* (cinnamate 4-hydroxylase) and *C3H* (p-coumarate 3-hydroxylase) genes was significantly upregulated (Figure 6c). This synergistic activation promotes three critical sequential conversions: cinnamic acid to p-coumaric acid, cinnamoyl-CoA to p-coumaroyl-CoA, and p-coumaroyl-CoA to feruloyl-CoA. Conversely, the *PRX* (peroxidase) gene family exhibited expression divergence: two members were upregulated, while six members were significantly downregulated (Figure 6d). This differential expression likely redirects the metabolic flux governing lignin monomer polymerization. These coordinated changes demonstrate that *GhSTZ* precisely modulates rate-limiting steps within these metabolic pathways, thereby influencing the synthesis of defense compounds.

### 2.7. Validation of Transcriptome Data via RT-qPCR

To assess the reliability of the transcriptome data and the proposed molecular mechanisms, the expression levels of 18 key genes involved in pentose and glucuronate interconversions, phenylpropanoid biosynthesis, and other signaling pathways were validated using RT-qPCR.

In the pentose and glucuronate interconversion pathway, *PLY* (Ghi_D02G08696) expression was significantly upregulated by 4.2-fold. In contrast, *PLY* (Ghi_D11G07386) and *PLY* (Ghi_A11G07641) expression decreased to 0.3-fold and 0.2-fold of the control levels, respectively. Furthermore, the expression of Ghi_A01G08156, Ghi_D06G03566, Ghi_A07G00931, Ghi_A12G07981, Ghi_A02G02236, and Ghi_A09G15106 was significantly upregulated, with increases ranging from 1.7- to 5.5-fold (Figure 7a).

Genes associated with phenylpropanoid metabolism were generally upregulated in *GhSTZ*-silenced plants. Specifically, the expression of *PRX* (Ghi_D03G00291), *PRX* (Ghi_D05G13146), and *PRX* (Ghi_A05G11436) increased by 2.1-fold, 6.5-fold, and 10.5-fold, respectively. Additionally, the expression of Ghi_A06G00066, Ghi_A12G09521, and Ghi_D12G15581 was significantly elevated, showing increases of 8.5- to 16.5-fold (Figure 7b).

Within the plant–pathogen recognition pathway, the expression of the cyclic nucleotide-gated channel genes *CNGC* (Ghi_D09G00616) and 3-ketoacyl-CoA synthase genes *KCS* (Ghi_A05G07196) increased by 2.7-fold and 13.5-fold, respectively. The expression of *KCS* (Ghi_D10G08036) was also significantly upregulated by 7.8-fold (Figure 7c).

## 3. Discussion

Despite decades of concerted efforts in breeding and widespread adoption of insect- and disease-resistant cotton varieties in China, VW remains a pivotal constraint on cotton production [28,29]. The co-evolutionary dynamics between the pathogen and its host necessitate an in-depth dissection of disease resistance mechanisms [30,31,32]. Previous research has predominantly focused on positive regulators, leaving the roles of negative regulators inadequately understood [33,34]. This study identified the C2H2 ZFP transcription factor *GhSTZ* as a novel negative regulator (Figure 3). Its domain architecture shares similarity with other stress-responsive C2H2 zinc finger proteins (Figure 2a,c). Through silencing *GhSTZ*, we reveal for the first time that it compromises cotton resistance via a tripartite inhibitory mechanism: suppression of PPP-driven metabolic reprogramming, repression of phenylpropanoid metabolism-mediated lignin deposition, and negative regulation of plant–pathogen recognition-associated immune signaling. This discovery provides a new perspective for understanding the negative regulatory network of plant immunity.

The altered expression of key PPP enzyme genes induced by *GhSTZ* silencing led to glucose accumulation (Figure 4) and may potentially result in NADPH buildup. This not only supplies reducing power for ROS scavenging systems like the glutathione cycle but also provides carbon skeletons for the phenylpropanoid pathway [35]. The significant enhancement of lignin deposition was directly associated with the activation of phenylpropanoid pathway genes C4H and C3H (Figure 6c). The divergent expression within the *PRX* gene family likely optimizes the spatial distribution of lignin polymerization by modulating peroxidase activity. This finding partially parallels the mechanism in *Nicotiana tabacum* where *TaMyb1D* reduces resistance by suppressing phenylpropanoid metabolism [36]. Notably, pathogens often hijack host PPP metabolites like pentoses for proliferation [37], and high *GhSTZ* expression may exacerbate this metabolic vulnerability (Figure 2d).

The optimized ROS homeostasis in *GhSTZ*-silenced plants is particularly crucial. DAB staining indicated moderate ROS accumulation (Figure 4a), while CAT activity was concurrently elevated (Figure 4b), demonstrating a balance between burst intensity and scavenging efficiency. Moderate ROS not only activates NLR (Nucleotide-binding domain Leucine-rich Repeat)-mediated ETI (Effector-Triggered Immunity) immunity [38] but can also enhance PRR (Pattern Recognition Receptor) receptor activity through oxidative modification [22,39]. Conversely, high *GhSTZ* expression may lead to NADPH deficiency and excessive ROS accumulation, thereby disrupting immune signaling and creating a microenvironment that favors pathogen colonization. The dual role of glucose in plant defense is as follows: In GhSTZ-silenced plants, elevated glucose levels potentially activate defense genes via the sugar-enhanced resistance model. At the same time, the downregulation of *PME* and *PG1-pec* in the PPP reduces cell wall pectin degradation, limiting carbon source acquisition by pathogens. This aligns with findings that overexpression of *CWIN* enhances disease resistance in rice and sugar induces *PR* gene expression in Arabidopsis [40,41,42]. However, the naturally high expression of *GhSTZ* restricts sugar accumulation by suppressing the PPP, simultaneously weakening sugar signaling while increasing the availability of carbon sources to the pathogen. Specifically, the activation of key PPP genes in silenced plants likely promotes NADPH generation, enhancing ROS scavenging and lignin synthesis. Concurrently, the upregulated expression of receptor-like kinases and pathogenesis-related proteins indicates enhanced immune signaling. Thus, *GhSTZ* may both reinforce defense through glucose accumulation while simultaneously exacerbating disease by providing carbon sources for pathogens.

The strategy suppressing *GhSTZ* to prioritize resource allocation to defense metabolism rather than passively restricting nutritional competition provides a novel concept for disease resistance breeding: targeted editing of negative regulators can break the vicious cycle of metabolic suppression–immune compromise. While this study elucidates the regulatory framework of the *GhSTZ*–PPP–phenylpropanoid–immune network, key questions remain: Does *GhSTZ* directly bind to the promoters of hub genes like G6PDH and PAL? Is its function dependent on the JA/ET signaling pathway? Do pathogen effectors target *GhSTZ* to manipulate host metabolism? Future investigations employing ChIP-seq, hormone mutant complementation, and effector screening will provide deeper insights. At the application level, generating *GhSTZ* knockout materials using CRISPR/Cas9 combined with field evaluation will accelerate its utilization in resistance breeding. This research not only expands the theory of negative regulation in plant immunity but also lays the foundation for designing multi-target disease resistance strategies that synergize metabolism and immunity.

## 4. Materials and Methods

### 4.1. Plant and Fungal Materials

The upland cotton (*Gossypium hirsutum*) line TM-1 was cultivated in a controlled growth chamber maintained at 25 °C, 70% relative humidity, and under a 16 h light/8 h dark photoperiod. The defoliating *V. dahliae* strain *Vd*991 was cultured on PDA (potato dextrose agar) solid medium at 25 °C. Mycelia were subsequently collected and transferred to Czapek’s liquid medium for approximately 5 days of incubation at 26 °C with shaking at 180 rpm. The final spore suspension concentration was adjusted to 1×10^7^ CFU/mL. The Laboratory of Crop Functional Genomics and Molecular Improvement, College of Life Sciences, Xinjiang Agricultural University, provided both the TM-1 cotton line and the *Vd*991 fungal strain.

### 4.2. Identification of Key Genes and Co-Expression Network Analysis

To identify key regulatory genes involved in the cotton response to *V. dahliae* infection, publicly available transcriptome data (FASTA format) from the NCBI Gene Expression Omnibus (GEO) database (GSE221845), representing cotton samples post-pathogen inoculation, were retrieved. The downloaded data underwent redundancy removal to ensure analytical quality. Subsequently, heatmaps were generated using R language (packages pheatmap or ggplot2) to visualize the expression patterns of *ZFP* gene family members. Hierarchical clustering was employed to group genes with similar expression profiles, with color gradients depicting expression level changes. Furthermore, WGCNA was performed using the R package WGCNA (v1.71) to construct gene co-expression networks. This analysis included data normalization, identification of co-expressed gene modules via hierarchical clustering and dynamic tree cutting, assessment of module stability, and correlation analysis of modules with target phenotypes to screen for hub gene modules significantly associated with cotton resistance to VW. The network structure of key gene modules was visualized and optimized using Cytoscape (v3.10.1).

### 4.3. Bioinformatics Analysis

The full-length sequence of *GhSTZ* (Gene ID: Gohir.D01G108400) was retrieved from the cotton functional genomics database phytozome (https://phytozome-next.jgi.doe.gov/) (accessed on 1 August 2024). The physicochemical properties of the encoded protein were predicted using the online tool ProtParam (https://web.expasy.org/protparam/) (accessed on 1 August 2024). Homologous protein sequences of GhSTZ were obtained from the NCBI database (Cytoscape). Phylogenetic tree construction was performed using MEGA11 software (X, Sudhir Kumar, Philadelphia, PA, USA). Conserved motif and domain analyses were conducted using MEME Suite (https://meme-suite.org/meme/) (accessed on 1 August 2024) and TBtools software (2.007, Chengjie Chen, Guangzhou, China).

### 4.4. Gene Expression Analysis

TM-1 cotton plants at the two-true-leaf stage were inoculated with *Vd*991 spore suspension 1 × 10^7^ CFU/mL using the root wounding method. Root, stem, and leaf tissues were sampled at 0, 0.5, 1, 2, 4, 8, 12, 24, and 48 hpi, flash-frozen in liquid nitrogen, and stored at −80 °C. Total RNA was extracted using the Plant Total RNA Extraction Kit RNAprep Pure (Tiangen, Beijing, China). Genomic DNA removal and cDNA synthesis were performed using the EasyScript^®^ One-Step gDNA Removal and cDNA Synthesis SuperMix (TransGen, Beijing, China). RT-qPCR was conducted using TransStart^®^ Top Green qPCR SuperMix (TransGen, Beijing, China). The *GhUBQ7* (LOC107912293) geneF:5′-GAAGGCATTCCACCTGACCAAC-3′, R:5′-CTTGACCTTCTTCTTCTTGTGCTTG-3′ served as the internal reference gene. The relative expression level of *GhSTZ* was calculated using the 2^−ΔΔCT^ method. Three technical replicates were performed for each sample.

### 4.5. VIGS in Cotton

A specific fragment targeting the *GhSTZ* gene forward primers 5′-GAATTCACTTTCAGGAGCCAACGATG-3′ and reverse primers 5′-GGATCCAATTTTGGGTGGCCTTTTGC-3′ were designed using the SGN-VIGS website (https://vigs.solgenomics.net/) (359 bp). The amplified fragment was ligated into the *Eco*RI/*BamH*I-digested pTRV2 vector using seamless cloning and transformed into DH5α competent cells. The recombinant plasmid pTRV2::*GhSTZ* was then transformed into *Agrobacterium* GV3101. Controls included pTRV2::*GhCLA1* positive control and pTRV2::*00* empty vector control. *Agrobacterium* cultures OD_600_ = 1.0 harboring pTRV2::*GhSTZ* or the control plasmids were infiltrated into the cotyledons of cotton seedlings at the fully expanded cotyledon stage. Plants were kept in darkness for 24 h before transfer to normal light conditions. Each treatment group consisted of 30 seedlings and was repeated three times.

### 4.6. Disease Resistance Phenotyping and Pathogen Quantification

When the positive control pTRV2::*GhCLA1* plants exhibited the albino phenotype approximately 14 days post-infiltration (dpi), RT-qPCR was used to confirm *GhSTZ* silencing efficiency using *GhUBQ7* as the internal reference gene with three technical replicates. Successfully silenced plants and control plants were inoculated with *Vd*991 spore suspension 1×10^7^ CFU/mL using the root wounding method [43,44]. Disease severity was evaluated 20 dpi according to the Chinese National Standard “Technical Specification for Evaluation of Cotton Resistance to Diseases and Insect Pests Part 5: Verticillium Wilt GB/T 22101.5-2009”, using the following scale: Grade 0 asymptomatic, Grade 1 1–25% leaves wilted, Grade 2 26–50% leaves wilted, Grade 3 51–75% leaves wilted, and Grade 4 76–100% leaves wilted or plant dead. Simultaneously, the relative abundance of *V. dahliae* DNA within stems was quantified by RT-qPCR using the pathogen-specific primers ITS-F: AAAGTTTTAATGGTTCGCTAAGA and VE1-R: CTTGGTCATTTAGAGGAAGTAA, with *GhUBQ7* as the internal reference gene. Vascular bundle browning was assessed visually after stem dissection.

### 4.7. Physiological and Biochemical Parameter Assays

Fresh leaf tissue (0.1 g) from differentially treated cotton plants was ground in liquid nitrogen and leaf glucose content was determined using a Glucose Content Assay Kit (Solarbio, Beijing, China), with a standard curve (0–100 μg·mL^−1^ glucose) used for calculation; stem tissue from differentially treated plants was sectioned using a cryo-microtome, lignin deposition was observed under an optical microscope using a Lignin Staining Solution Kit (Phygene, Fuzhou, China), and lignin content was measured using a Lignin Content Assay Kit (Solarbio, Beijing, China); intact true leaves from differentially treated plants were stained with 3,3′-diaminobenzidine (DAB) using the Plant Tissue Reactive Oxygen Species Detection Kit (Solarbio, Beijing, China) for histochemical detection; fresh leaf tissue (0.1 g) from differentially treated plants was ground in liquid nitrogen and relative CAT enzyme activity was calculated using a CAT Activity Assay Kit (Solarbio, Beijing, China); three biological replicates were performed for each treatment.

### 4.8. Transcriptome Sequencing and Analysis

Transcriptome sequencing was performed on stem tissues collected from pTRV2::*GhSTZ* and pTRV2::*00* plants at 24 h following inoculation with *V. dahliae* strain *Vd*991. Total RNA extraction preceded library preparation for Illumina NovaSeq 6000 sequencing. Raw sequencing data underwent quality control via Trimmomatic to eliminate low-quality sequences scoring below Phred 20 and adapter contaminants, yielding clean reads. Subsequent alignment to the *Gossypium hirsutum* reference genome v3.1 employed HISAT2, with gene expression quantification as Fragments Per Kilobase of transcript per Million mapped reads (FPKM) values performed using StringTie. Differentially expressed genes were identified based on thresholds of absolute log_2_ fold change ≥1 and false discovery rate <0.05. Functional interpretation included Gene Ontology and Kyoto Encyclopedia of Genes and Genomes enrichment analyses alongside protein–protein interaction network construction, all executed through ClusterProfiler. All phenotyped and transcriptome-sampled plants were from the same growth batch to avoid batch effects.

### 4.9. RT-qPCR Validation of Disease Resistance-Related Genes

Validation of disease resistance-related gene expression was conducted via RT-qPCR using stem tissues harvested from pTRV2::*GhSTZ* and pTRV2::*00* plants at 24 hpi-*Vd*991 inoculation. Target genes spanned three defense pathways: pentose–glucuronate interconversion (Ghi_D02G08696, Ghi_D11G07386, Ghi_A11G07641, Ghi_A01G08156, Ghi_D06G03566, Ghi_A07G00931, Ghi_A12G07981, Ghi_A02G02236, and Ghi_A09G15106), phenylpropanoid biosynthesis (Ghi_D03G00291, Ghi_D05G13146, Ghi_A05G11436, Ghi_A06G00066, Ghi_A12G09521, and Ghi_D12G15581), and plant–pathogen recognition (Ghi_D09G00616, Ghi_A05G07196, and Ghi_D10G08036). The RT-qPCR methodology mirrored Section 4.4 utilizing *GhUBQ7* for normalization. Three biological replicates per gene underwent analysis, with statistical significance between experimental and control groups assessed via Student’s *t*-test at *p* < 0.05.

## 5. Conclusions

This study identifies *GhSTZ* as a critical negative regulator of cotton resistance to VW. Silencing *GhSTZ* significantly enhances disease resistance by coordinately activating three defense mechanisms: metabolic reprogramming via the pentose phosphate pathway, increasing NADPH/glucose for redox balance; lignin deposition through phenylpropanoid biosynthesis, fortifying physical barriers; and immune recognition signaling. Then, *GhSTZ*-mediated suppression of metabolic–immune crosstalk creates vulnerabilities exploited by *V. dahliae*. Targeting *GhSTZ* via gene editing presents a promising strategy for breeding wilt-resistant cotton by derepressing these synergistic defense pathways.

## Figures and Tables

**Figure 1 plants-14-02638-f001:**
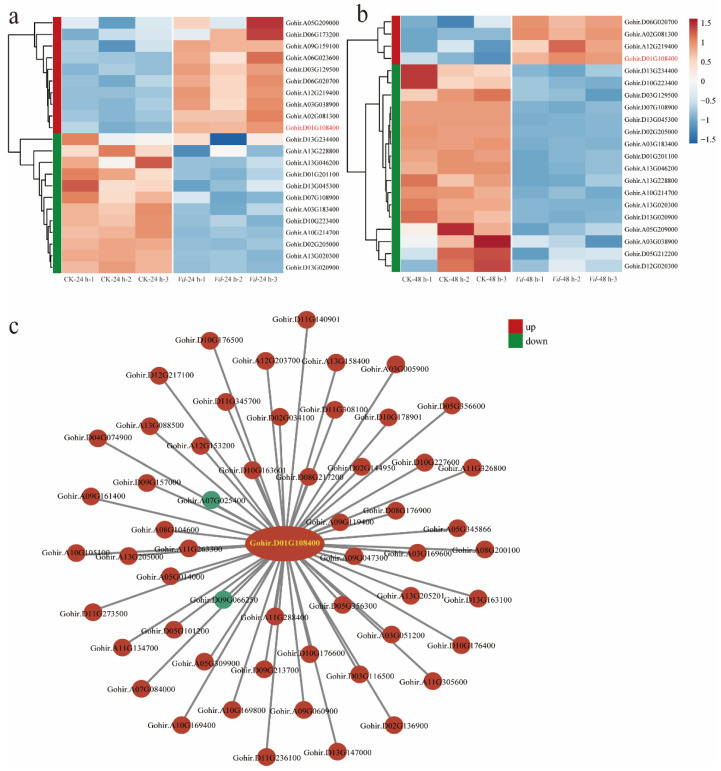
Analysis of differentially expressed genes in cotton before and after *V. dahliae* infection. (**a**) Expression analysis of *ZFP* genes 24 hpi. (**b**) Expression analysis of *ZFP* family genes 48 hpi. (**c**) WGCNA analysis of *ZFP* family genes.

**Figure 2 plants-14-02638-f002:**
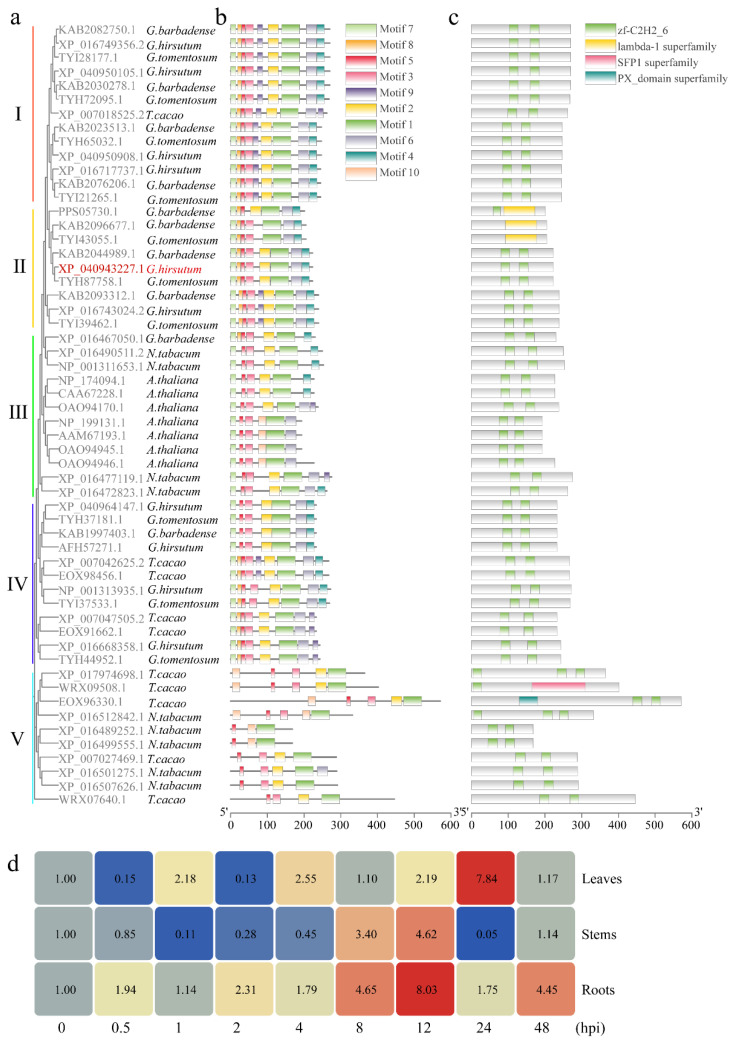
Bioinformatic and expression analysis of *GhSTZ*. (**a**) Maximum likelihood phylogenetic tree of *GhSTZ* orthologs (bootstrap = 1000). *Gossypium hirsutum* (*G. hirsutum*), *Gossypium barbadense* (*G. barbadense*), *Gossypium tomentosum* (*G. tomentosum*), *Nicotiana tabacum* (*N. tabacum*), *Arabidopsis thaliana* (*A. thaliana*), and *Theobroma cacao* (*T. cacao*). (**b**) Distribution of conserved motifs in *GhSTZ* orthologs. Colored boxes represent distinct motifs; zinc finger domains correspond to specific motifs. (**c**) Predicted conserved domains of GhSTZ orthologs highlighting C2H2 zinc finger domains. (**d**) Expression pattern analysis of *GhSTZ* in cotton plants following inoculation with VW. A heatmap analysis of *GhSTZ* gene expression across various organs (roots, stems, and leaves) of TM-1 cotton plants was conducted at multiple time points (0, 0.5, 1, 2, 4, 8, 12, 24, and 48 hpi) following inoculation with *V. dahliae*.

**Figure 3 plants-14-02638-f003:**
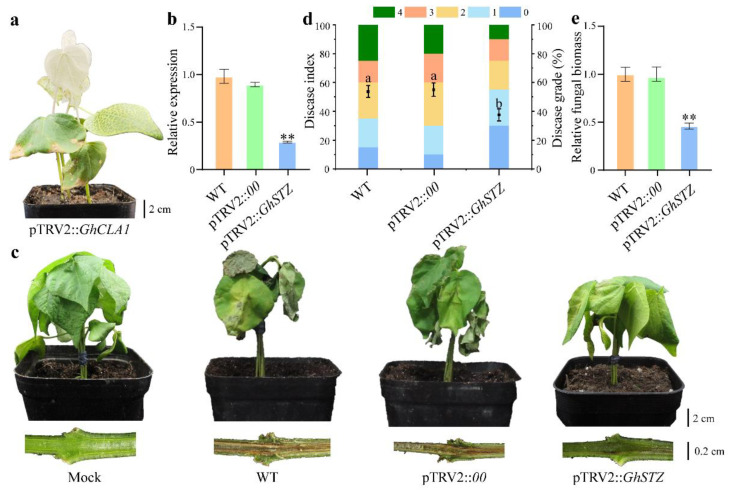
Functional validation of *GhSTZ* in VW defense. (**a**) Phenotype of cotton plants infiltrated with pTRV2::*GhCLA1* at 14 dpi. (**b**) Relative *GhSTZ* expression levels in WT, pTRV2::*00*, and pTRV2::*GhSTZ* plants by RT-qPCR. (**c**) Representative images showing leaf wilting and vascular browning in WT, pTRV2::*00*, and pTRV2::*GhSTZ* plants at 20 dpi. (**d**) Disease indices of inoculated plants were calculated based on leaf symptom grading. (**e**) Relative fungal biomass quantification in stems at 20 dpi. Data are presented as the mean ± standard error (*n* = 3). Statistical significance is indicated as follows: ** *p* < 0.01. Different lowercase letters indicate statistically significant differences (*p* < 0.05).

**Figure 4 plants-14-02638-f004:**
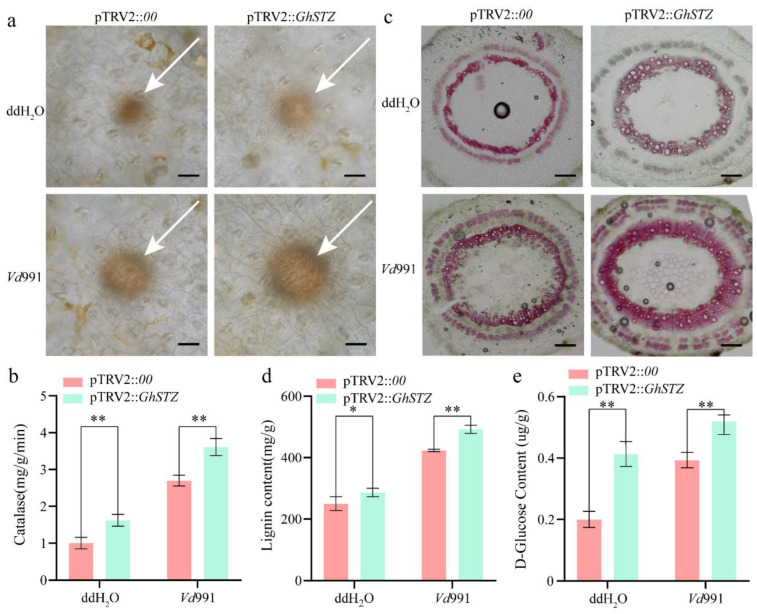
Physiological analysis of defense compound accumulation in *GhSTZ*-silenced plants. (**a**) Detection of ROS accumulation. DAB staining showing ROS accumulation (brown spots) in leaves of pTRV2::*GhSTZ* and pTRV2::*00* plants under water-treated (Mock) and *V. dahliae Vd*991-inoculated conditions. Scale bar = 100 μm. The arrow indicates the site of ROS accumulation. (**b**) CAT enzyme activity. CAT activity in leaves of pTRV2::*GhSTZ* and pTRV2::*00* plants under Mock and *Vd*991 inoculation treatments. (**c**) Vascular bundle structure visualization. Transverse sections of stems from pTRV2::*GhSTZ* and pTRV2::*00* plants stained to reveal vascular bundle morphology. Scale bar = 100 μm. (**d**) Lignin deposition and quantification. Upper panel: phloroglucinol staining of stem sections showing lignin accumulation (red signal) in pTRV2::*GhSTZ* and pTRV2::*00* plants. Scale bar = 100 μm. Lower panel: quantitative measurement of lignin content in stems. (**e**) Glucose content measurement. Glucose levels in leaves of pTRV2::*GhSTZ* and pTRV2::*00* plants under Mock and *Vd*991 inoculation treatments. Data represent mean ± SD (*n* = 3). Statistical analysis was conducted using the *t*-test, with significance indicated as follows: * *p* < 0.05 and ** *p* < 0.01.

**Figure 5 plants-14-02638-f005:**
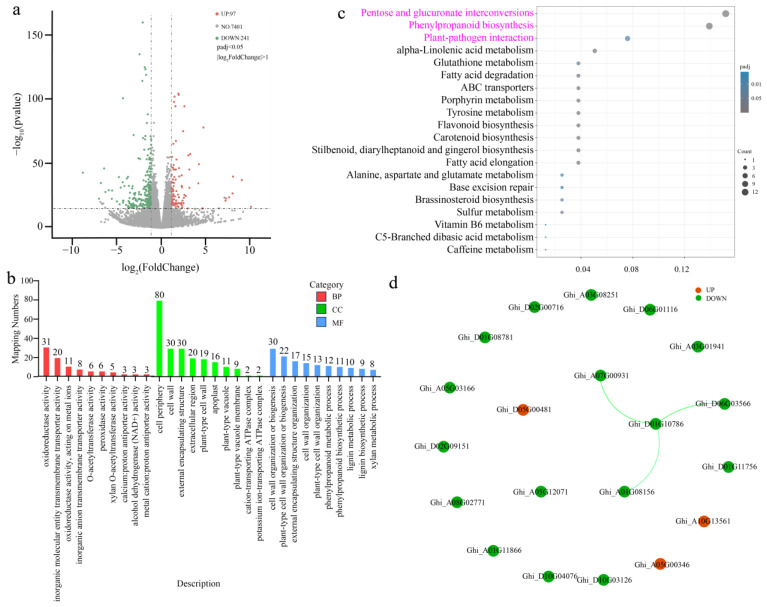
(**a**) Volcano plot of DEGs. Red dots represent significantly upregulated genes, green dots represent downregulated genes, and gray dots represent genes with no significant change in expression. (**b**) GO enrichment analysis of DEGs in pTRV2::*00* vs. pTRV2::*GhSTZ* cotton plants. Significantly enriched terms are shown within the categories biological process (BP), cellular component (CC), and molecular function (MF). (**c**) KEGG pathway enrichment analysis of DEGs is shown, with the pathway highlighted in pink representing the one containing the most enriched genes. (**d**) PPI network analysis of DEGs. Nodes represent proteins encoded by DEGs; edges represent predicted interactions—red nodes: upregulated genes; green nodes: downregulated genes.

**Figure 6 plants-14-02638-f006:**
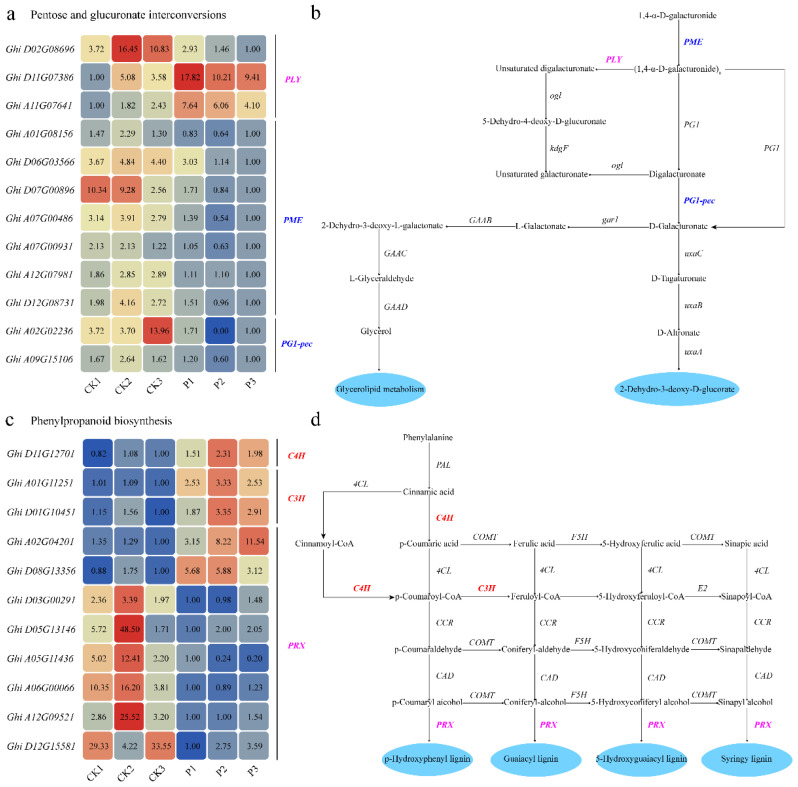
Regulation of metabolic pathway genes by *GhSTZ* silencing. (**a**) Expression changes of key genes in the pentose and glucuronate interconversion pathway. Gene names are colored to indicate expression change in pTRV2::*GhSTZ* plants relative to pTRV2::*00* controls. Purple: bidirectional regulation (significantly upregulated/downregulated context-dependently); blue: significantly downregulated. (**b**) Schematic of the pentose and glucuronate interconversion pathway. Arrows indicate metabolic reactions. (**c**) Expression changes of key genes in the phenylpropanoid biosynthesis pathway. Gene names are colored to indicate expression change: Red: significantly upregulated; purple: bidirectional regulation (significantly upregulated/downregulated context-dependently). (**d**) Schematic of the phenylpropanoid biosynthesis pathway. Arrows represent metabolic reactions.

**Figure 7 plants-14-02638-f007:**
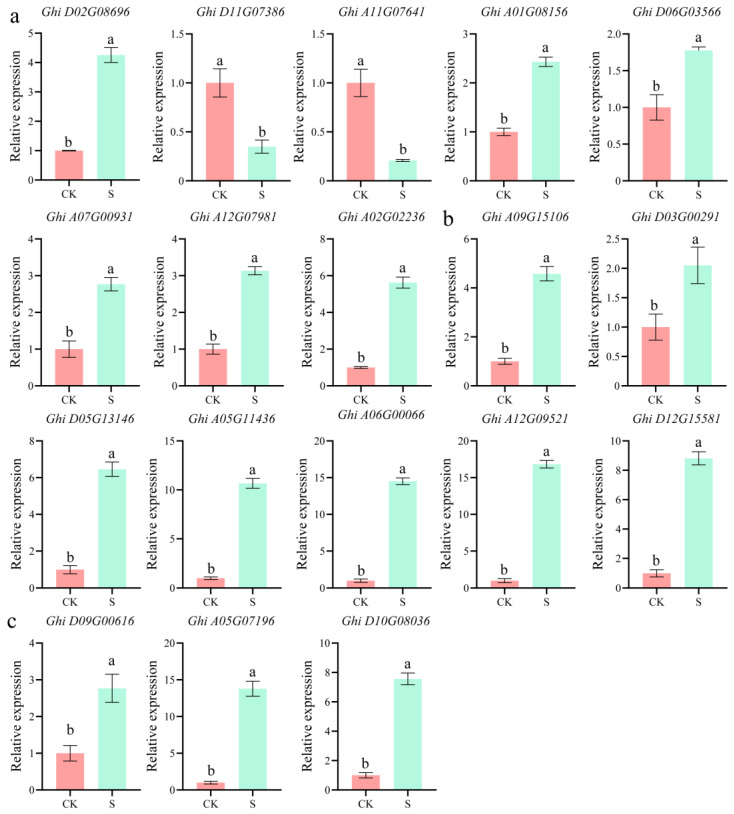
RT-qPCR validation of key defense-related gene expression in *GhSTZ*-silenced plants. (**a**) Expression levels of genes involved in the pentose and glucuronate interconversion pathway. (**b**) Expression levels of genes involved in the phenylpropanoid biosynthesis pathway. (**c**) Expression levels of genes involved in the plant–pathogen recognition pathway. pTRV2::*00* plants served as the control (CK), while pTRV2::*GhSTZ* plants represented the silenced group (S). Data are presented as mean ± SD (*n* = 3 biological replicates). Different lowercase letters indicate statistically significant differences (*p* < 0.05).

## Data Availability

The original contributions presented in this study are included in the article. Further inquiries can be directed to the corresponding author.

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
