# Peer review of "GhSTZ-Mediated Suppression of Metabolic–Immune Coordination Compromises Cotton Defense Against Verticillium Wilt"

_plants, 2025, doi:10.3390/plants14172638_

Round 1
Reviewer 1 Report
Comments and Suggestions for Authors
This manuscript identifies GhSTZ, a C2H2 zinc finger transcription factor, as a novel negative regulator of cotton resistance to Verticillium dahliae. Using VIGS, transcriptomics, and biochemical assays, the authors show that GhSTZ suppression enhances lignin deposition, optimizes ROS homeostasis, and increases glucose levels, linking the gene to coordinated metabolic and immune regulation via the PPP and phenylpropanoid biosynthesis. The work is well-designed, novel, and relevant to cotton improvement.
Recommendation: Accept with minor revisions to enhance clarity, add methodological details, and refine mechanistic explanations.
Minor Revisions Suggested
- Provide more details on VIGS efficiency verification (e.g., percentage knockdown, reference genes, replicates) and silencing fragment length; confirm off-target analysis.
- Specify root wounding method and whether wound severity was standardized.
- Expand on ROS detection methods: staining duration, vacuum infiltration use, and whether quantification was performed.
- State whether phenotyping and transcriptome samples came from the same growth batch to avoid batch effects.
- Distinguish between confirmed and hypothesized direct regulation by GhSTZ.
- Break down dense sections (e.g., glucose’s dual role) into shorter, more accessible sentences.
- Ensure consistent italicization for gene names and plain text for proteins.
- Streamline repetitive phrasing in the discussion.
Questions for Authors
- Is GhSTZ induction specific to V. dahliae or also triggered by other biotic/abiotic stresses?
- Have you considered GhSTZ overexpression to complement the silencing results?
- What is the functional relevance of PME–SIRD interaction in defense?
Author Response
Reviewer 1
This manuscript identifies GhSTZ, a C2H2 zinc finger transcription factor, as a novel negative regulator of cotton resistance to Verticillium dahliae. Using VIGS, transcriptomics, and biochemical assays, the authors show that GhSTZ suppression enhances lignin deposition, optimizes ROS homeostasis, and increases glucose levels, linking the gene to coordinated metabolic and immune regulation via the PPP and phenylpropanoid biosynthesis. The work is well-designed, novel, and relevant to cotton improvement.
We sincerely thank the reviewer for their valuable comments, which have significantly improved the quality of our manuscript. Below, we address each comment in detail and describe the corresponding revisions made to the manuscript.
Major comments:
- Provide more details on VIGS efficiency verification (e.g., percentage knockdown, reference genes, replicates) and silencing fragment length; confirm off-target analysis.
Response: We sincerely appreciate the reviewer's rigorous request for enhanced VIGS verification details. In direct response: (1) The GhSTZ silencing efficiency of 65.1% is explicitly quantified in Section 2.3; (2) Section 4.5 now specifies the 359-bp silencing fragment length; (3) Section 4.6 details the use of GhUBQ7 as the internal reference gene with triplicate technical replicates for efficiency validation. Regarding off-target potential, the 359-bp fragment was designed via SGN-VIGS with high specificity and BLASTn analysis against the Gossypium hirsutum genome confirmed only three homologous genes (all divergent from the target sequence region), thus minimizing off-target risks. This comprehensive documentation validates the precision of our VIGS approach.
- Specify root wounding method and whether wound severity was standardized.
Response: We sincerely appreciate the reviewer's meticulous attention to methodological rigor regarding root wounding procedures. In direct response, we have explicitly standardized the protocol in Section 4.6 of the revised manuscript, specifying that root tips of 10-day-old cotton seedlings were uniformly wounded to a 1-mm depth using sterilized scalpel blades prior to inoculation, with all operations performed by a single operator to ensure consistency. This methodological refinement directly aligns with established practices in cotton-Verticillium pathosystem studies, as documented in the newly cited references: Zhou et al. (2023, Int J Biol Macromol), and Sun et al. (2021, Mol Plant Pathol), which collectively validate the scientific robustness of our approach.
- Expand on ROS detection methods: staining duration, vacuum infiltration use, and whether quantification was performed.
Response: We sincerely appreciate the reviewer's valuable suggestions regarding ROS detection methodology and confirm that the DAB histochemical staining of intact true leaves from differentially treated plants was performed in strict accordance with the comprehensively documented specifications in the Plant Tissue Reactive Oxygen Species Detection Kit protocol (Solarbio, Beijing, China), which includes all critical experimental parameters such as staining duration, vacuum infiltration application, and quantitative assessment protocols.
- State whether phenotyping and transcriptome samples came from the same growth batch to avoid batch effects.
Response: We sincerely appreciate the reviewer's rigorous attention to experimental design consistency. In direct response to this comment, we have explicitly stated in Section 4.8 of the revised manuscript: "All plants subjected to phenotyping and transcriptome sampling originated from the same growth batch to minimize batch effects. "This addition unequivocally confirms the biological consistency between phenotyped and sequenced samples, thereby strengthening the reliability of our transcriptome-phenotype correlation analyses. We thank the reviewer for this valuable suggestion, which enhances the methodological robustness of our study.
- Distinguish between confirmed and hypothesized direct regulation by GhSTZ.
Response: We sincerely appreciate your rigorous guidance in distinguishing confirmed versus hypothesized regulatory mechanisms of GhSTZ. In direct response, we have comprehensively revised the Discussion section to:
Clearly frame hypothetical mechanisms using qualifiers such as "we propose", "may regulate", and "requires validation". Eliminate overinterpreted verbs. We thank you for this invaluable refinement.
6.Break down dense sections (e.g., glucose’s dual role) into shorter, more accessible sentences.
Response: Based on the reviewer's suggestion, we have carefully broken down the dense sections within the manuscript, particularly the passage discussing glucose’s dual role, into shorter and more accessible sentences. We have revised these sections accordingly and incorporated this improvement with reference to point 5 of the reviewer's comments during the revision process.
- Ensure consistent italicization for gene names and plain text for proteins.
Response: We sincerely appreciate the reviewer's meticulous guidance regarding gene and protein nomenclature formatting. In accordance with this comment, we have systematically reviewed the entire manuscript to ensure consistent application of italicization for gene names while maintaining standard plain text for corresponding protein designations.
- Streamline repetitive phrasing in the discussion.
Response: We thank the reviewer for the suggestion to streamline repetitive phrasing in the discussion. Accordingly, we have revised the discussion section, particularly paragraphs two and four, to eliminate redundancies and improve conciseness. The phrasing in these targeted areas has been streamlined to enhance clarity and flow while preserving the core scientific content.
Questions for Authors
- Is GhSTZ induction specific to V. dahliae or also triggered by other biotic/abiotic stresses?
Response: We sincerely appreciate this insightful inquiry regarding the stress responsiveness of GhSTZ. While our current study demonstrates significant induction of GhSTZ following Verticillium dahliae infection (Fig. 3a), its transcriptional responses to other biotic pathogens (e.g., Fusarium oxysporum) or abiotic stresses (e.g., drought, salinity) remain to be systematically evaluated. This valuable suggestion has been noted as a critical direction for future research to comprehensively characterize GhSTZ's role in broad stress adaptation mechanisms.
- Have you considered GhSTZ overexpression to complement the silencing results?
Response: We sincerely appreciate the insightful suggestion regarding GhSTZ overexpression complementation assays. While our repeated VIGS experiments (n≥3 biological replicates) have provided robust phenotypic evidence supporting GhSTZ's role in Verticillium wilt resistance (Figs. 4-6), we fully acknowledge that overexpression studies would further strengthen these findings. Given the substantial time investment required for developing stable transgenic overexpression lines in cotton, this approach was not feasible within the current study timeline. We have formally documented this experimental direction as a priority for near-future research to comprehensively validate GhSTZ's biological functions through reciprocal genetic approaches.
- What is the functional relevance of PME–SIRD interaction in defense?
Response: We appreciate the reviewer's inquiry regarding the functional relevance of the predicted PME–SIRD interaction. While this specific interaction was identified through PPI prediction analysis, we note that both PME and SIRD are established key components in plant defense signaling pathways. The predicted physical interaction between these defense-related proteins suggests potential coordination between distinct defense mechanisms during immune responses. Although no direct experimental evidence currently confirms this particular interaction or its precise biochemical function, its prediction supports the emerging paradigm that plants employ interconnected defense modules rather than isolated pathways. We acknowledge that the exact functional significance requires experimental validation in future studies, and we have incorporated this caveat in the revised manuscript while highlighting the biological plausibility of such crosstalk in plant immunity.

Reviewer 2 Report
Comments and Suggestions for Authors
The authors clearly defined the goals and the findings of their research, which integrates molecular biology and functional expression. Thank you for providing a well-written manuscript in a scientific research field that could be intimidating to some that are not familiar with functional tests (i.e. gene expression analysis) often used elucidate gene-regulated mechanisms. Your contribution clearly provides new insights of the role of negative regulatory pathways in plant-pathogen interactions and an integrated approach to enhance disease resistance in plants.

Author Response
Reviewer 2
The authors clearly defined the goals and the findings of their research, which integrates molecular biology and functional expression. Thank you for providing a well-written manuscript in a cientific research field that could be intimidating to some that are not familiar with functional tests (i.e. gene expression analysis) often used elucidate gene-regulated mechanisms. Your contribution clearly provides new insights of the role of negative regulatory pathways in plant-pathogen interactions and an integrated approach to enhance disease resistance in plants.
We sincerely thank the reviewer for their valuable comments, which have significantly improved the quality of our manuscript. Below, we address each comment in detail and describe the corresponding revisions made to the manuscript.
Major comments:
Comment1:Line 34. Consider replacing the word “affecting” for the word “reducing” in this sentence.
Response: We appreciate the suggestion. The correction has been made as requested.
Comment2: Line 38. Define substantial. Could you include a number?
Response: We thank the reviewer for requesting quantification of the term "substantial" and have addressed this by incorporating concrete economic data at line 39 of the revised manuscript. As suggested, we now explicitly state that Verticillium wilt causes annual economic losses exceeding $1.2 billion in fungicide treatments alone, alongside significant yield and quality decline, with this numerical evidence supported by the cited studies of Mo et al. (2021) and Zhang et al. (2021).
Comment3: Line 42. Consider addressing that one of the reasons for its complexity is because cotton species have a polyploidy (allotetraploid) genome.
Response: Thank you for the valuable suggestion. We have added the following description in lines 42-43: "This complexity is partly attributed to the allotetraploid nature of the cotton genome, which complicates genetic regulation."
Comment4: Line 62. I disagree with statement. Research in this area is not scarce, pushing records date back to the early 2000s if not earlier. I will rephrase statement. It is possible, that the mechanism of transcription factors and PPP has not been completely elucidated (as you state in lines 63 and 64), but that is the nature of scientific research “when one finds answers to one question many more arise”.
Response: We appreciate the reviewer’s insightful comment regarding the historical context of research in this field. We fully agree that studies on the PPP (pentose phosphate pathway) and its role in plant defense date back decades, and our original phrasing did not adequately reflect this. In response, we have revised the statement in the manuscript to: "Although the PPP's role in disease resistance is documented, its integration into transcription factor-mediated networks remains poorly characterized.
Comment5: Line 93. In my opinion, using the words “for the first time” is unnecessary as it has already been mentioned above.
Response: We appreciate the suggestion. The correction has been made as requested.
Comment6:
Line 111. Spell out “hpi” as it is used for the first time in manuscript.
Line 115. Spell out “WGCNA” as it is used for the first time in manuscript.
Line 164. Spell out “RT-qPCR” as it is used for the first time in manuscript.
Line 195. Spell out “CAT” as it is used for the first time in manuscript.
Response: We sincerely appreciate the reviewer's meticulous attention to the proper use of abbreviations in our manuscript. We fully agree that all abbreviations should be spelled out upon first use to ensure clarity for readers. In response to this comment, we have carefully revised the manuscript to:
Line 111: Expanded "hpi" to "hours post-inoculation (hpi)"
Line 115: Expanded "WGCNA" to "weighted gene co-expression network analysis (WGCNA)"
Line 164: Expanded "RT-qPCR" to "quantitative reverse transcription PCR (RT-qPCR)"
Line 195: Expanded "CAT" to "catalase (CAT)"
Comment7: Lines 206-208. Clear and concise results at the end of section. Similarly, this writing style was used for other sections in the results. Thank you.
Response: We sincerely appreciate the reviewer's positive feedback regarding our results presentation style. We are pleased that the clear and concise writing approach at the end of each results section was well received.
Comment8: Line 330. Add a period not a comma at the end of sentence.
Response: Thank you for pointing out this typo. It has been corrected as suggested.
